# Effects of Dietary Fiber Level and Forage Particle Size on Growth, Nutrient Digestion, Ruminal Fermentation, and Behavior of Weaned Holstein Calves under Heat Stress

**DOI:** 10.3390/ani14020275

**Published:** 2024-01-16

**Authors:** Mohammad-Hossein Izadbakhsh, Farzad Hashemzadeh, Masoud Alikhani, Gholam-Reza Ghorbani, Mohammad Khorvash, Mostafa Heidari, Morteza Hosseini Ghaffari, Farhad Ahmadi

**Affiliations:** 1Department of Animal Sciences, College of Agriculture, Isfahan University of Technology, Isfahan 84156-83111, Iranf.hashemzadeh@cc.iut.ac.ir (F.H.);; 2Institute of Animal Science, University of Bonn, 53115 Bonn, Germany; 3School of Agriculture, Food and Ecosystem Sciences, Faculty of Science, The University of Melbourne, Parkville, VIC 3010, Australia; 4Department of Eco-Friendly Livestock Science, Institute of Green Bio Science and Technology, Seoul National University, Pyeongchang 25354, Republic of Korea

**Keywords:** calf health, environmental temperature, sorting behavior, nutrient utilization

## Abstract

**Simple Summary:**

Global temperatures are on the rise, and this poses a significant threat to animal farming. However, there are certain nutritional strategies that can help mitigate the harmful effects of heat stress, particularly for the growing heifer calves who play a crucial role in the future profitability and sustainability of a dairy farm. In this study, growing calves experiencing heat stress consumed more feed and digested nutrients more efficiently when offered diets with moderate content of dietary fiber and shorter alfalfa hay particle size. Feeding a diet with a moderate level of fiber resulted in better growth, lower rectal temperature, and longer lying behavior, an indication of better animal comfort. Our findings highlight the importance of dietary fiber level and forage particle size in shaping calf performance and behavior under challenging environmental conditions.

**Abstract:**

This experiment was designed to investigate the effects of feeding diets with different fiber content and forage particle size on the performance, health, nutrient digestion, rumen fermentation, and behavioral and sorting activity of Holstein dairy calves kept under elevated environmental temperature. Sixty weaned Holstein female calves (age = 96.7 ± 7.62 days old; body weight = 82.4 ± 10.4 kg) were randomly assigned to one of 4 treatments arranged in a 2-by-2 factorial design in a 70-day experiment. Dietary forage content (moderate, 22.5%; or high, 40.0% on DM basis) and alfalfa hay particle size (short, 4.39 mm; or long, 7.22 mm as geometric mean) were the experimental factors, resulting in the following combinations: (1) high-fiber (HF) diets with forage-to-concentrate ratio of 40:60 and long particle-sized alfalfa hay (LPS; HF-LPS); (2) HF diets with short particle-sized alfalfa hay (SPS; HF-SPS); (3) moderate-fiber (MF) diets with forage-to-concentrate ratio of 22.5:77.5 with LPS (MF-LPS); and (4) MF diets with SPS (MF-SPS). The temperature–humidity index averaged 73.0 ± 1.86, indicating that weaned calves experienced a moderate extent of heat stress. Fiber level and AH particle size interacted and affected dry matter intake, with the greatest intake (4.83 kg/d) observed in MF-SPS-fed calves. Final body weight was greater in calves receiving MF vs. HF diets (164 vs. 152 kg; *p* < 0.01). Respiration rate decreased when SPS vs. LPS AH was included in HF but not MF diet. Lower rectal temperature was recorded in calves fed MF vs. HF diet. Digestibility of dry matter and crude protein was greater in calves fed MF than HF diets, resulting in lower ruminal pH (6.12 vs. 6.30; *p* = 0.03). Fiber digestibility was greater in calves fed SPS compared with those fed LPS alfalfa hay. Feeding HF compared with MF diet increased acetate but lowered propionate molar proportions. The inclusion of SPS vs. LPS alfalfa hay decreased lying time in HF diet (920 vs. 861 min; *p* < 0.01). Calves fed MF vs. HF diets spent less time eating but more time lying, which is likely indicative of better animal comfort. Dietary fiber level and forage particle size interacted and affected sorting against 19 mm particles, the extent of which was greater in HF-SPS diet. Overall, dietary fiber level had a stronger effect than forage particle size on the performance of weaned calves exposed to a moderate degree of heat stress as feeding MF vs. HF diet resulted in greater feed intake, final body weight, structural growth measures, nutrient digestion, as well as longer lying behavior. The inclusion of SPS alfalfa hay in MF diets increased feed consumption.

## 1. Introduction

Appropriate management strategies during pre- and post-weaning phases are crucial, laying the foundations for long-term productivity, behavior, and health of heifer calves, and ultimately contribute to a successful heifer-rearing program [1,2]. The rise in global temperature hampers the profitability and sustainability of animal production systems, a problem that is expected to worsen in the future [3,4]. Due to decreased metabolic heat production and more efficient heat dissipation mechanisms, growing calves are more resistant to thermal stress than mature cows [5]. However, there is evidence that heat stress may affect the calf physiology, productivity, rumen function, and welfare [6,7,8]. Newer findings suggest the long-term consequences of exposure to high environmental temperatures on the health and future productivity of young animals, including calves and growing heifers [9,10,11]. Calves kept in hot climates may experience reduced gastrointestinal motility and a series of physiological and metabolic reactions diverting energy towards thermoregulation, which is associated with decreased feed intake and growth efficiency [10]. The nutrient digestibility was greater in heifers exposed to heat stress, likely because of declined feed intake and increased retention time in the whole gastrointestinal tract [12]. However, other studies have not reported such responses [13,14].

There are certain dietary approaches that promote feed intake and growth rate in growing animals, which may help mitigate the harmful effects of heat stress. For example, the proper balance between forage and concentrate proportion in the diet of growing calves improved growth efficiency and promoted rumen development. The forage component of the diet promotes the muscular development of the rumen [15], while concentrate intake fuels growth [16] and stimulates rumen papillae formation [17]. However, the dietary fiber content must be balanced to ensure efficient rumen functioning and to minimize metabolic dysfunctions [5,10]. Feeding excessive amounts of readily fermentable carbohydrates might raise the risk of ruminal acidosis and parakeratosis, which can have negative implications on calf health, feed utilization efficiency, and gastrointestinal function [18,19]. Identifying the proper forage-to-concentrate proportion in the diet of growing calves ensures adequate rumen development and functionality, which is associated with improved performance.

Conflicting evidence exists in the literature about the proper forage-to-concentrate ratio in the diet of weaned calves. Several studies have reported that decreasing the forage-to-concentrate ratio in the calf diet increased feed intake [20,21,22]. For example, Karami et al. [23] have reported that weaned calves fed diets with a forage-to-concentrate ratio of 20:80 vs. 50:50 or 35:65 ratios from 70 to 120 days of age consumed more feed, grew faster, and were more efficient in converting feed nutrients toward growth. However, Namboothiri et al. [24] have reported that decreasing the forage-to-concentrate ratio in the diet of growing calves from 6 to 12 months of age had no effect on intake but improved growth performance.

The physical form of forage is another nutritional factor that may influence the rumen function and performance of young calves [25]. Providing a diet with appropriate forage particle size is necessary to promote chewing and salivation, which increases rumen pH and supports rumen health and proper function [26,27]. According to Montoro et al. [25], feeding coarsely chopped (3 to 4 cm) grass hay compared with finely chopped (2 mm) grass hay increased feed intake and nutrient digestibility during week 8 (the week after weaning) but decreased the time calves spent in non-nutritive oral behaviors. Nemati et al. [26] have reported that feeding alfalfa hay (AH) with medium particle size (geometric particle size = 3 mm) improved feed intake and growth of weaned calves (aged 51 to 70 days) compared with a diet containing AH with a geometric particle size of 1 mm. Increased consumption of diets containing AH with medium vs. fine particle size was attributed to increased capacity and muscle development of the reticulorumen [28,29,30]. In contrast, minor changes in rumen development and fermentation indicators were observed in dairy calves when straw particle length increased from 3.04 to 12.7 mm (as a geometric mean) [31]. The inconsistencies could possibly be due to the physical form of the starter feed, the amount of feed supplied, and the growth stage of the calves. Previous studies have not paid much attention to the potential effects of forage particle size on post-weaning calf nutrition, especially under heat-stress conditions. Therefore, further research is needed to determine the effects of forage particle size on intake and growth, and this research aims to fill this knowledge gap and highlight the importance of forage particle size as a factor when formulating diets for post-weaned calves.

To promote optimal nutrition in weaned calves, it is important to formulate diets to provide sufficient levels of physically effective fiber, as concentrate intake increases sharply after weaning. We hypothesized that feeding a diet with moderate fiber content and long AH particles would improve growth performance and increase nutrient digestion compared with a diet with high fiber content and long AH particles. Therefore, our objective was to investigate the interaction between dietary fiber content (medium or high) and AH particle size (long or short) on performance, nutrient digestion, rumen fermentation, and the behavior of weaned calves under elevated environmental temperatures.

## 2. Materials and Methods

Prior to the beginning of the study, all procedures involving animals were reviewed and approved by the Animal Care and Use Committee of the Isfahan University of Technology, and the animal-involved methods were conducted in accordance with the regulations of the Iranian Council of Animal Care (1995). The experiment was carried out at a commercial dairy farm (Namfar) located in Isfahan, Iran. The trial was conducted from June to September 2021. Humidity and air temperature in the stall area were recorded daily using a hygrometer/thermometer (model: HTC-1; Zhejiang Junkaishun Industrial and Trade Co., Ltd., Jinhua, China). Temperature–humidity index (THI) was calculated using the following equation [32]: THI = 0.8 × maximum T + (minimum relative humidity/100) × (maximum T − 14.4) + 46.4, where T is the temperature (°C).

### 2.1. Animals, Management, and Dietary Treatments

#### 2.1.1. Pre-Trial Handling of Calves

After birth, Holstein calves were separated from their mothers, weighed, and kept in separate indoor pens with bedding made of wheat straw. Colostrum was fed using a plastic bottle with a nipple attachment after being heat-treated for 60 min at 60 °C. The calves were kept in indoor pens for five days, and on days 2 and 3, their second and third milkings after calving (at 6:00 a.m. and 3:00 p.m.), they were given transition milk. Then, whole pasteurized milk was fed for the next two days. On their sixth day of life, calves were placed in separate outdoor hutches (1.3 m × 2.5 m) and kept there until day 70. The hutches were bedded with wood shaving and wheat straw. Whole milk was provided using an aluminum bucket at an approximate temperature of 38 °C with the following composition: 11.7 ± 0.18% total solids, 2.92 ± 0.09% protein, and 3.33 ± 0.12% fat (mean ± standard deviation). Milk was offered three times a day at 6:00 a.m., 3:00 p.m., and 8:00 p.m. from day 6 to 30 (9 L/d), twice a day at 6:00 a.m. and 3:00 p.m. from day 31 to 44 (8 L/d) and 45 to 65 (5 L/d), and then once a day (only morning, 6:00 a.m.) from day 66 to 70 (3 L/d). Weaning occurred on day 70. Animals had unlimited access to starter feed and fresh water. The starter feed included 5% forage (50% AH and 50% wheat straw) and 95% concentrate mix (5% ground barley, 55.17% ground corn, 30% soybean meal, 3% whole soybean, 1% fat powder, 1% mineral supplement, 1% vitamin supplement, 1.5% calcium carbonate, 1.5% sodium bicarbonate, 0.3% dicalcium phosphate, 0.5% salt, and 0.03% monensin).

#### 2.1.2. Trial Handling of Calves

Sixty weaned Holstein female calves (mean age = 96.7 ± 7.62 days old) were weighed (mean body weight = 82.4 ± 10.4 kg) and, after balancing for age and body weight, were moved to twelve group pens with five calves housed in each pen (3.0 m × 6.0 m) that were in turn each bedded with sawdust. Manure was collected and fresh bedding was added to each pen daily. Each pen was randomly allocated into 1 of 4 treatments, as follows: (1) high-fiber (HF) diets with forage-to-concentrate ratio of 40:60 with long particle-sized AH (LPS; HF-LPS); (2) HF diets with short particle-sized AH (SPS; HF-SPS); (3) moderate-fiber (MF) diets with forage-to-concentrate ratio of 22.5:77.5 with LPS AH (MF-LPS); and (4) MF diets with SPS AH (MF-SPS). Second-cut AH was harvested at 10% flowering, wilted, and baled (small rectangular bales). Subsequently, AH was chopped with a forage harvester (Golchin Trasher Hay Co., Isfahan, Iran) to obtain the long and short particle sizes. The chemical composition and particle size distribution of corn silage and AH are presented in Table 1 and Table 2, respectively. The ingredients and chemical compositions of the experimental diets are listed in Table 3 and Table 4, respectively. Diets were formulated with CPM-Dairy version 3.0 software (Cornell University, University of Pennsylvania, and the Miner Institute). Animals were fed once daily at 8:00 a.m. Each day, concentrate and forage portions were thoroughly mixed and an appropriate amount in the form of a total mixed ration was manually weighed. Fresh water and diet were freely available to all calves. According to the orts in the previous day, the total amount of feed supplied to each pen was adjusted to provide 5–10% refusals.

### 2.2. Sampling and Laboratory Analyses

Approximately 2 kg of each forage, including corn silage and AH, and weekly samples of feed and refusal of each pen were collected on two consecutive days of each week and stored at −20 °C until further analysis. The samples were mixed thoroughly, dried at 55 °C for 48 h in a forced-air oven, and then ground to pass through a 1 mm screen. Chemical compositional analysis of ground samples was made using the AOAC [33] standard methods: ether extract (AOAC International 920.39), N (AOAC International 990.03), and ash (AOAC International 942.05). Analyses of neutral detergent fiber (NDF) and acid detergent fiber (ADF) were performed using ANKOM^220^ Fiber Analyzer (ANKOM Technology, Macedon, NY, USA). The NDF procedure was inclusive of heat-stable α-amylase and residual ash. Non-fiber carbohydrates (NFC) were calculated as NFC = 100 − (NDF + crude protein + ether extract + ash). Forage, feed, and refusal samples were sieved to determine the particle size distribution. The Penn State Particle Separator, which consists of 19, 8, and 1.18 mm diameter sieves and a pan, was used to monitor the particle size distribution. Geometric mean particle length (GMPL) was calculated in accordance with the ASAE [34].

Daily feed intake for each group was measured by weighing the amount of feed supplied and refused. Dry matter intake (DMI; kg/d) was then calculated on a pen basis. Individual body weight measurements were made before morning feeding at 0, 10, 25, 40, 55, and 70 days of the study using an electronic scale. Body weights recorded for each calf were used to calculate individual average daily gain (ADG). Feed conversion ratio was calculated as average daily feed intake/ADG of the pen. Skeletal growth measures, including withers height, hip height, hip width, belly girth, heart girth, and body length, were measured before morning feeding at 0, 10, 25, 40, 55, and 70 days of the study, as described by Khan et al. [35]. Calves were evaluated weekly for body condition score (BCS) using a 5-point scale [36].

Respiratory rate (breaths/min) and rectal temperature and were recorded thrice each week. The frequency of movement of the abdominal muscles in the flank during respiration was quantified visually to determine the respiratory rate and was expressed as breaths/min. Rectal temperature was determined between 2:00 and 3:00 p.m. with the insertion of a thermometer (PIC Vedodigit II, Pic Solution Co., Como, Italy) directly into the rectum for 1 min. Calf health was monitored daily during the experiments. The University of Wisconsin Calf Health Scoring chart [37] was utilized to visually assess and score the health status of animals. For the ear and eye score categories, a score of 0 indicates a normal, healthy animal.

To determine nutrient digestibility, fecal samples were taken directly from the calf rectum 3 h after morning feeding on 5 consecutive days in weeks 5 and 10. Fecal samples were kept frozen (−20 °C) until analysis. The samples were later thawed overnight and dried at 55 °C for 72 h in a forced-air oven. These samples were ground to pass through a 1 mm screen and analyzed for nutrients, as described earlier. Acid-insoluble ash was used as an internal marker to measure the apparent total-tract digestibility of nutrients [38].

During the trial, rumen fluid was collected twice—on day 35 and day 70—4 h after the morning feeding. The collection was undertaken by using a stomach tube attached to a vacuum pump. To prepare the fluid for analysis of VFA via gas chromatography, 4 mL of the rumen fluid, filtered through four layers of cheesecloth, was mixed with 1 mL of 25% metaphosphoric acid. VFA analysis was undertaken using CP-9002 gas chromatography (Chrompack, Middelburg, The Netherlands). The chromatography used a 50 m (0.32 mm i.d.) silica-fused column (CP-Wax Chrompack Capillary Column by Varian, Palo Alto, CA, USA) with crotonic acid (1:7, *v*/*v*) as an internal standard, as described by Hashemzadeh-Cigari et al. [39].

Behavioral activities, including eating, rumination, lying, standing drinking, and nonnutritive oral behavior, were recorded at d 34 and 69 of the study for 24 h starting from 8:00 a.m., as detailed previously [40]. Four trained researchers who were not aware of the treatments were employed to record the behavioral activities. Sorting index was computed as a fraction of actual intake to expected intake of particles retained on each sieve [41].

### 2.3. Statistical Analysis

The MIXED PROC in SAS (2003, version 9.4, SAS Institute Inc., Cary, NC, USA) was used for data analysis. The outcome assessed varied depending on whether the observational unit was the pen or the individual calf. The model included the random effect of individual animal as the observational unit for calf body weight, ADG, structural growth measures, body condition score, health status parameters, nutrient digestion, rumen fermentation parameters, and feeding behaviors. The model included the random effect of the pen as the experimental unit for nutrient intake and feed conversion ratio, based on the experimental design and statistical models for pen studies described by St-Pierre [42], Tempelman [43], and Bello et al. [44]. Time served as a repeated measure for parameters including nutrient intake, performance, structural growth, nutrient digestion, rumen fermentation, feeding behaviors and sorting index, and health parameters. The model included fixed effects of fiber content, AH particle size, sampling time, and their interactions. An assessment was conducted on three variance–covariance structures (autoregressive type 1, Toeplitz, and compound symmetry), and, upon considering the Bayesian information criterion, the autoregressive type 1 covariance structure was identified as the most fitting choice. Initial calf measurements, including body weight and skeletal growth measures, were considered covariates for the analysis of body weight and skeletal growth data. The normality of all residuals was assessed using the Shapiro–Wilk statistic via the UNIVARIATE PROC of SAS [45]. Additionally, the homogeneity of variance was examined using Levene’s test and visually evaluated through quantile–quantile plots. Where the data did not adhere to the assumptions of normality, a log transformation (base 10) was applied. The Tukey–Kramer adjustment was used for means comparison. Significance was established at a threshold of *p* ≤ 0.05, while trends were identified within the range of 0.05 < *p* ≤ 0.10.

**Table 1 animals-14-00275-t001:** Chemical composition (±SD) of corn silage and alfalfa hay.

Chemical Composition, % of DM	Corn Silage	Alfalfa Hay
DM, %	25.4 ± 0.16	97.7 ± 0.22
Organic matter	93.4 ± 0.37	88.6 ± 0.32
Crude protein	7.20 ± 0.04	13.8 ± 0.42
Crude protein	2.41 ± 0.31	1.82 ± 0.28
NDF	53.4 ± 0.91	46.2 ± 1.10
NDF	32.3 ± 0.54	37.8 ± 0.22

**Table 2 animals-14-00275-t002:** Physical characteristics of corn silage and alfalfa hay.

Items	Corn Silage	Alfalfa Hay Particle Size		SEM	*p* Value ^2^
		Long	Short		
% DM retained on sieve ^1^					
19 mm	22.9 ± 4.42	20.9	3.70	1.42	<0.01
8 mm	61.1 ± 2.22	27.8	33.4	1.66	0.02
1.18 mm	14.9 ± 0.93	38.6	45.3	1.58	<0.01
<1.18 mm (pan)	1.10 ± 1.77	12.7	17.6	2.32	0.14
peNDF_8_	44.8 ± 1.38	22.5	17.1	1.16	<0.01
peNDF_1.18_	52.8 ± 0.98	40.2	38.0	1.07	0.14
GMPL ^3^, mm	12.0 ± 0.95	7.22	4.39	0.41	<0.01

^1^ Particle size distribution was made using the Penn State Particle Separator [46]. PeNDF denotes physically effective NDF. PeNDF_8_ and peNDF_1.18_ were determined as multiplication of dietary NDF content by pef_8_ and pef_1.18_, respectively. Pef_8_ and pef_1.18_ are physical effectiveness factors as proportions of particles retained on 2 [47] and 3 sieves [46], respectively. ^2^
*p* value showing the level of significance between short and long alfalfa hay. ^3^ GMPL = geometric mean particle length. SEM = standard error of the means.

**Table 3 animals-14-00275-t003:** Ingredients of the experimental diets.

Ingredients, % of DM	Dietary Fiber Content	
	Moderate Fiber	High Fiber
Alfalfa hay	18.0	32.0
Corn silage	4.50	8.00
Corn grain, ground	16.5	12.38
Barley grain, ground	26.6	19.88
Soybean meal	14.25	14.25
Wheat bran	15.63	8.00
Fat powder ^1^	0.25	1.63
Vitamin supplement ^2^	0.75	0.75
Mineral supplement ^3^	0.75	0.75
Salt	0.25	0.25
Calcium carbonate	1.25	0.88
Bicarbonate	1.00	1.00
Magnesium oxide	0.25	0.25

^1^ Fat powder contained 99.5% total fat, 40% C16:0, 50% C18:0, and 10% C18:1. ^2^ Contained (per kg): vitamin A = 1,300,000 IU, vitamin D_3_ = 360,000 IU, vitamin E = 12,000 IU. ^3^ Contained (per kg): zinc = 16 g, manganese = 10 g, copper = 4 g, iodine = 0.15 g, cobalt = 0.12 g, iron = 0.8 g, and selenium = 0.08 mg.

**Table 4 animals-14-00275-t004:** Chemical composition and physical characteristics of experimental diets differing in fiber level (F) and alfalfa hay particle size (PS) *.

Items	Moderate Fiber		High Fiber		SEM	*p* Values		
	LPS	SPS	LPS	SPS		F	PS	F × PS
Chemical composition								
DM, % as fed	53.6	52.7	53.4	53.3	0.96	0.83	0.33	0.28
Organic matter, % of DM	90.9	90.6	90.4	90.9	0.69	0.47	0.88	0.71
Crude protein, % of DM	16.4	16.3	15.9	16.2	0.46	0.58	0.82	0.76
Ether extract, % of DM	3.11	3.20	4.82	5.01	0.36	<0.01	<0.01	<0.01
NDF, % of DM	28.9	28.5	33.6	32.4	0.24	<0.01	<0.01	<0.01
ADF, % of DM	13.9	13.2	18.7	17.5	0.21	<0.01	<0.01	<0.01
Forage NDF, % of DM	10.7	10.7	19.0	19.0	–	–	–	–
NFC ^1^, % of DM	42.4	43.0	35.8	36.4	–	–	–	–
ME, Mcal/kg DM	2.17	2.17	2.16	2.16	–	–	–	–
% DM retained on sieve ^2^								
19 mm	6.4 ^b^	2.4 ^c^	11.3 ^a^	3.6 ^c^	0.48	<0.01	<0.01	<0.01
8 mm	15.3 ^d^	17.6 ^c^	20.3 ^b^	25.2 ^a^	0.40	<0.01	<0.01	<0.01
1.18 mm	60.6	62.9	51.6	55.0	0.79	<0.01	<0.01	0.29
<1.18 mm (pan)	17.7	17.1	16.8	16.2	0.61	0.20	0.37	0.82
peNDF_8_	6.35	5.75	10.61	9.47	0.20	<0.01	<0.01	0.17
peNDF_1.18_	23.9	23.8	27.8	27.9	0.19	<0.01	0.42	0.97
GMPL ^3^ (mm)	3.91	3.71	4.66	4.23	0.07	<0.01	<0.01	0.10

* The dietary treatments were as follows: (1) high-fiber (HF) diets with forage-to-concentrate ratio of 40:60 with long particle-sized alfalfa hay (LPS; HF-LPS); (2) HF diets with short particle-sized alfalfa hay (SPS; HF-SPS); (3) moderate-fiber (MF) diets with forage-to-concentrate ratio of 22.5:77.5 with LPS alfalfa hay (MF-LPS); and (4) MF diets with SPS alfalfa hay (MF-SPS). ^1^ Nonfibrous carbohydrate = 100 − (crude protein + NDF + ether extract + ash). ^2^ Particle size distribution was made using the Penn State Particle Separator [46]. PeNDF denotes physically effective NDF. PeNDF_8_ and peNDF_1.18_ were determined as multiplications of dietary NDF content by pef_8_ and pef_1.18_, respectively. Pef_8_ and pef_1.18_ denote physical effectiveness factors as proportions of particles retained on 2 [47] and 3 sieves [46], respectively. ^3^ GMPL = geometric mean particle length. SEM = standard error of the means; ^a–d^ Within a row, least-square means with dissimilar superscripts differ (*p* ≤ 0.05).

## 3. Results

### 3.1. Meteorological Data

Throughout the experiment, the mean, maximum, and minimum THI values averaged 73.0 ± 1.86, 86.2 ± 2.79, and 62.4 ± 1.79, respectively (Figure 1).

### 3.2. Nutrient Composition and Particle Size Distribution

A large difference was observed in the particle size distribution of AH with both sizes. LPS AH had more particles retained on 19 mm screens, less particles retained on the 8.0 and 1.18 mm screens, and similar amounts of particles shorter than the 1.18 mm compared with SPS AH. GMPL and peNDF_8_ were greater for LPS vs. SPS AH (Table 2).

Chemical composition and particle size distributions of dietary treatments are listed in Table 4. There were interactions between fiber level and AH particle size on particles retained on 19 and 8 mm sieves. Particles retained on the 19 mm screen followed the decreasing order of HF-LPS > MF-LPS > HF-SPS = MF-SPS. Particles retained on the 8 mm screen were in the order of HF-SPS > HF-LPS > MF-SPS > MF-LPS. HF vs. MF diet resulted in a lesser proportion of particles retained on the 1.18 mm sieve (61.8 vs. 53.1%, *p* < 0.01) but greater GMPL (4.45 vs. 3.81 mm; *p* < 0.01), the top two sieves (30.2 vs. 20.9%; *p* < 0.01), and peNDF_8_ (10.0 vs. 6.0% of DM, *p* < 0.01). Diets containing SPS AH had a greater proportion of particles retained on the 1.18 mm sieve but had shorter GMPL (3.97 vs. 4.30 mm; *p* < 0.01), lower particles retained on the top two sieves, and peNDF_8_ (7.61 vs. 8.48% of DM; *p* < 0.01) compared with diets containing LPS AH.

### 3.3. Performance Responses

Nutrient intakes, weight gain, feed conversion ratio, skeletal growth measures, and BCS results are listed in Table 5. Calf body weight, ADG, and DM intake at different time points (days 10, 25, 40, 55, and 70 of the experiment) are also presented in Figure 2, Figure 3 and Figure 4, respectively. Fiber levels and AH particle size interacted and affected DM, NDF, and NFC intakes, with the greatest values of DM and NFC observed in calves fed the MF-SPS diet. Forage NDF intake was greater in calves receiving HF than in MF diets (0.79 vs. 0.48 kg/d; *p* < 0.01). Calves fed MF vs. HF diet had a greater final body weight (164 vs. 152 kg; *p* < 0.01). There was a fiber level by time interaction for body weight, with calves fed the MF diets having a greater body weight than those fed HF diet from day 40 to 70 of the trial. Further, calves fed MF diets had greater ADG (1.19 vs. 1.05 kg/d; *p* < 0.01) than those fed HF diets throughout the study. Calves fed MF vs. HF diet tended to be more efficient in converting feed nutrients toward growth (3.84 vs. 3.94; *p* = 0.08).

**Table 5 animals-14-00275-t005:** Nutrient intake, performance, and structural growth measures of weaned calves fed diets differing in fiber level (F) and alfalfa hay particle size (PS) *.

Items	Moderate Fiber		High Fiber		SEM	*p* Values						
	LPS	SPS	LPS	SPS		F	PS	F × PS	Time (T)	F × T	PS × T	F × PS × T
Total DM intake, kg/d	4.31	4.83	4.29	4.01	0.18	0.03	0.43	0.04	<0.01	0.70	0.97	0.91
NDF intake, kg/d	1.24	1.37	1.44	1.31	0.06	0.24	0.97	0.02	<0.01	0.91	0.98	0.90
NFC intake, kg/d	1.82	2.08	1.53	1.47	0.07	<0.01	0.19	0.03	<0.01	0.31	0.97	0.94
Forage NDF intake, kg/d	0.46	0.51	0.81	0.77	0.03	<0.01	0.92	0.09	<0.01	0.36	0.97	0.96
Final body weight, kg	162.9	164.1	154.1	149.6	3.56	<0.01	0.65	0.44	–	–	–	–
ADG, kg/d	1.17	1.21	1.07	1.03	0.05	<0.01	0.94	0.42	<0.01	0.30	0.45	0.30
Feed conversion ratio ^1^	3.68	3.99	4.00	3.88	0.31	0.08	0.22	0.79	<0.01	0.61	0.99	0.38
Structural growth measures												
Wither height, cm	100.2	100.3	99.7	99.2	0.27	<0.01	0.63	0.16	<0.01	0.55	0.86	0.89
Hip height, cm	101.9	102.6	102.2	101.5	0.28	0.13	0.81	0.01	<0.01	0.96	0.93	0.89
Hip width, cm	23.5	23.3	23.3	23.1	0.13	0.35	0.19	0.95	<0.01	0.10	0.75	0.96
Belly girth, cm	133.0	132.7	131.8	131.2	0.48	0.01	0.33	0.74	<0.01	0.21	0.84	0.97
Heart girth, cm	113.7	114.2	112.7	111.0	0.33	<0.01	0.04	<0.01	<0.01	0.11	0.99	0.94
Body length, cm	109.0	109.9	110.2	107.9	0.57	0.23	0.07	<0.01	<0.01	0.97	0.96	0.94
Body condition score	3.43	3.57	3.47	3.32	0.02	<0.01	0.98	<0.01	<0.01	0.59	0.87	0.91

* The dietary treatments were as follows: (1) high-fiber (HF) diets with forage-to-concentrate ratio of 40:60 with long particle-sized alfalfa hay (LPS; HF-LPS); (2) HF diets with short particle-sized alfalfa hay (SPS; HF-SPS); (3) moderate-fiber (MF) diets with forage-to-concentrate ratio of 22.5:77.5 with LPS alfalfa hay (MF-LPS); and (4) MF diets with SPS alfalfa hay (MF-SPS); NFC = non-fibrous carbohydrates. ^1^ Feed conversion ratio = total DM intake/ADG. SEM = standard error of mean.

Dietary fiber level and AH particle size interacted and affected heart girth and body length, with the smallest values observed in HF-SPS-fed calves. Calves offered MF diets had greater wither height, belly girth, and heart girth compared with calves fed HF diets. Fiber level and AH particle size interacted and affected BCS, with calves fed LPS AH having greater BCS compared with those fed SPS AH in HF diets, though an inverse response was observed with MF diets.

### 3.4. Health Status Indicators

Respiration rate, rectal temperature, and eye/ear score as indicators of calf health are presented in Table 6. Weekly data on rectal temperature and respiration rate are also presented in Figure 5 and Figure 6, respectively. Interactions existed between fiber level and forage particle size for respiration rate (*p* < 0.01), as feeding LPS vs. SPS AH increased respiration rate only in HF but not MF diets. Feeding HF vs. MF diets increased rectal temperature (38.99 vs. 39.04 °C; *p* = 0.01) and ear score (*p* < 0.01), while an opposite effect was noted for eye score. Respiration rate increased (*p* = 0.01) when calves were fed LPS vs. SPS AH.

**Table 6 animals-14-00275-t006:** Health status indicators of weaned calves fed diets differing in fiber level (F) and alfalfa hay particle size (PS) *.

Items	Moderate Fiber		High Fiber		SEM	*p* Values						
	LPS	SPS	LPS	SPS		F	PS	F × PS	Time (T)	F × T	PS × T	F × PS × T
Respiration rate, breaths/min	59.8	60.4	61.4	56.9	0.761	0.20	0.01	<0.01	<0.01	0.58	0.91	0.98
Rectal temperature, °C	38.99	38.98	39.02	39.06	0.019	0.01	0.42	0.22	<0.01	0.50	0.89	0.41
Eye score	0.587	0.537	0.463	0.489	0.029	<0.01	0.67	0.18	<0.01	0.96	0.69	0.72
Ear score	0.371	0.314	0.404	0.491	0.029	<0.01	0.59	0.01	<0.01	0.60	0.99	0.06

* The dietary treatments were as follows: (1) high-fiber (HF) diets with forage-to-concentrate ratio of 40:60 with long particle-sized alfalfa hay (LPS; HF-LPS); (2) HF diets with short particle-sized alfalfa hay (SPS; HF-SPS); (3) moderate-fiber (MF) diets with forage-to-concentrate ratio of 22.5:77.5 with LPS alfalfa hay (MF-LPS); and (4) MF diets with SPS alfalfa hay (MF-SPS). SEM = standard error of the means.

### 3.5. Nutrient Digestibility and Rumen Fermentation

Nutrient digestibility and rumen fermentation variables are listed in Table 7. Nutrient digestibility remained unaffected by the interactive effect of dietary fiber level and AH particle size. Digestibility of DM, organic matter, and crude protein increased when calves were fed the MF diet rather than the HF diet. NDF and ADF digestibility were greater in calves fed SPS compared with those fed LPS AH.

**Table 7 animals-14-00275-t007:** Nutrient digestion and rumen fermentation characteristics of weaned calves fed diets differing in fiber level (F) and forage particle size (PS) *.

Items	Moderate Fiber		High Fiber		SEM	*p* Values						
	LPS	SPS	LPS	SPS		F	PS	F × PS	Time (T)	F × T	PS × T	F × PS × T
Digestibility, %												
DM	72.2	77.1	68.6	69.8	2.62	0.03	0.19	0.40	0.60	0.37	0.20	0.42
Organic matter	74.2	78.8	71.2	72.0	2.11	0.04	0.24	0.38	0.59	0.41	0.22	0.40
Crude protein	72.8	77.4	68.9	66.6	2.77	0.02	0.69	0.27	0.91	0.87	0.31	0.46
NDF	49.4	60.3	49.4	53.6	2.19	0.12	<0.01	0.12	0.88	0.20	0.31	0.23
ADF	44.8	52.6	48.8	51.4	1.30	0.39	0.01	0.13	0.91	0.11	0.39	0.32
Rumen fermentation parameters												
pH	6.05	6.19	6.20	6.39	0.08	0.03	0.06	0.89	0.08	0.69	0.37	0.82
Ammonia-N, mg/dL	10.7	10.3	10.8	10.4	1.18	0.94	0.69	0.98	0.24	0.47	0.30	0.84
Total VFA, mM	83.8	82.5	85.6	85.0	4.03	0.60	0.82	0.92	0.83	0.14	0.71	0.91
Individual VFA, mol/100 mol												
Acetate	56.4	57.3	64.7	66.7	0.92	<0.01	0.13	0.55	0.29	0.73	0.48	0.73
Propionate	30.7	30.1	22.6	20.3	1.16	<0.01	0.23	0.44	0.33	0.44	0.57	0.66
Butyrate	9.68	9.44	9.84	9.96	0.35	0.35	0.82	0.58	0.76	0.52	0.61	0.63
Isobutyrate	0.55	0.48	0.55	0.65	0.04	0.03	0.71	0.02	0.12	0.08	0.29	0.43
Valerate	1.73	1.74	1.43	1.29	0.19	0.04	0.72	0.68	0.80	0.31	0.18	0.96
Isovalerate	0.94	0.91	0.92	1.12	0.08	0.26	0.34	0.16	0.12	0.09	0.83	0.99
Acetate: propionate	1.96	1.99	2.93	3.33	0.13	<0.01	0.09	0.15	0.22	0.67	0.55	0.95

* The dietary treatments were as follows: (1) high-fiber (HF) diets with forage-to-concentrate ratio of 40:60 with long particle-sized alfalfa hay (LPS; HF-LPS); (2) HF diets with short particle-sized alfalfa hay (SPS; HF-SPS); (3) moderate-fiber (MF) diets with forage-to-concentrate ratio of 22.5:77.5 with LPS alfalfa hay (MF-LPS); and (4) MF diets with SPS alfalfa hay (MF-SPS). SEM = standard error of the means.

There were no interactions between fiber level and AH particle size for rumen fermentation variables except for isobutyrate molar proportion, which was greatest in calves fed the HF-SPS diet. Calves fed HF diets had higher rumen pH compared with those fed MF diets. Ruminal pH tended to be greater in calves fed diets containing SPS AH than those fed LPS AH. Main dietary factors or their interaction had no effect on ruminal total VFA and ammonia-N concentrations. Feeding HF vs. MF diets resulted in greater acetate (65.7 vs. 56.9; *p* < 0.01), but lower propionate molar proportions (21.5 vs. 30.4; *p* < 0.01), which resulted in a greater acetate-to-propionate ratio. Valerate molar proportion also decreased (1.36 vs. 1.74; *p* = 0.04) when calves received HF rather than MF diets.

### 3.6. Behavioral Activities

Data on feeding behavior and sorting activity are presented in Table 8. An interaction between fiber content and particle size of AH was observed for standing (*p* < 0.01) and lying behaviors (*p* < 0.01); calves fed LPS AH spent more time standing with those fed SPS AH in MF diets, though an inverse response was observed with HF diets. Feeding SPS vs. LPS AH decreased lying behavior only in HF but not MF diets. Feeding SPS vs. LPS AH resulted in less lying time only in the HF diet. There was a time-by-AH particle size effect for drinking behavior, with calves fed SPS AH spending more time to drink compared with those fed LPS AH only on day 70 of the study. Additionally, calves receiving SPS AH spent more time eating on day 35 of the study, but this response was not observed on day 70 of the study. Calves fed MF vs. HF diets spent less time eating (205 vs. 275 min; *p* < 0.01) and ruminating (106 vs. 130 min; *p* < 0.01), but more time lying (953 vs. 891 min; *p* < 0.01).

**Table 8 animals-14-00275-t008:** Feeding behaviors and sorting index of weaned calves fed diets differing in fiber level (F) and forage particle size (PS) *.

Items	Moderate Fiber		High Fiber		SEM	*p* Values						
	LPS	SPS	LPS	SPS		F	PS	F × PS	Time (T)	F × T	PS × T	F × PS × T
Feeding behavior												
Eating, min	198	211	262	288	5.53	<0.01	0.01	0.44	<0.01	0.80	<0.01	0.18
Rumination, min	103	109	126	133	2.77	<0.01	0.02	0.83	0.02	0.98	0.02	0.58
Lying, min	944	962	920	861	10.5	<0.01	0.05	<0.01	<0.01	0.51	0.97	0.22
Standing, min	250	209	207	240	9.23	0.54	0.64	<0.01	<0.01	0.93	0.65	0.77
Drinking, min	13.8	16.3	15.3	18.9	1.74	0.24	0.08	0.75	<0.01	0.40	0.02	0.04
NNOB, min	34.3	42.5	35.4	32.5	3.74	0.23	0.48	0.14	0.51	0.04	<0.01	0.37
Sorting index ^1^												
19 mm	95.2 *	97.8	90.5 *	84.5 *	1.93	<0.01	0.37	0.02	0.01	0.40	0.15	0.87
8 mm	103 *	102 *	96.5 *	98.6 *	0.98	<0.01	0.61	0.05	0.56	0.19	0.88	0.86
1.18 mm	103 *	101 *	103 *	102 *	0.54	0.48	0.01	0.59	<0.01	0.99	0.09	0.97
<1.18 mm (pan)	93.7 *	93.6 *	101 *	97.6 *	1.18	<0.01	0.10	0.14	<0.01	0.52	0.60	0.98

* The dietary treatments were as follows: (1) high-fiber (HF) diets with forage-to-concentrate ratio of 40:60 with long particle-sized alfalfa hay (LPS; HF-LPS); (2) HF diets with short particle-sized alfalfa hay (SPS; HF-SPS); (3) moderate-fiber (MF) diets with forage-to-concentrate ratio of 22.5:77.5 with LPS alfalfa hay (MF-LPS); and (4) MF diets with SPS alfalfa hay (MF-SPS); NNOB = non-nutritive oral behavior, defined as oral activities such as licking, consuming non-eatable items, tongue rolling, and grooming, excluding eating or drinking behavior. ^1^ No sorting is indicated by a value equal to 100, sorting for is indicated by a value greater than 100, and sorting against is indicated by a value of less than 100. * *p* < 0.05; sorting values differ from 100. SEM = standard error of the means.

Fiber level and AH particle size of AH interacted (*p* = 0.02) and affected the sorting activity for long particles, as the extent of sorting against 19 mm particles was greatest in calves fed HF-SPS. All calves sorted against the 19 mm particles, the extent of which was greater in calves fed the HF than the MF diet. Regardless of AH particle size, calves fed MF sorted in favor of particles retained on the 8 mm sieve, but those fed the HF diet sorted against these particles (102.3 vs. 97.7; *p* < 0.01). All calves sorted in favor of particles retained on the 1.18 mm sieve, the extent of which was generally greater in calves fed LPS AH than in those fed SPS AH. All calves sorted against particles finer than 1.18 mm (pan) except those fed HF-LPS, which sorted for these particles.

## 4. Discussion

Studies on dairy cows have shown that an average THI above the threshold of 68 is a signal for the initiation of heat stress [48,49]. Rumination time or milk yield may decrease when THI reaches above this threshold. However, no criterion has been established for young calves, perhaps because it is impossible to measure performance parameters such as milk yield in growing calves [50]. Because of their greater surface area relative to body weight and lower heat production than dairy cows, calves may be better equipped to withstand warmer temperatures [5,51]. However, growing calves still suffer from heat stress as it can affect their physiology, growth, feed intake, rumen function, and wellbeing [6,7,8]. According to Dado-Senn et al. [50], THI in calves raised in a shaded, subtropical environment is the best indicator of heat stress, and it is important to keep a regular check on the calves when the THI exceeds 65 to 69.

Compared with adult cows, young calves have a narrower thermoneutral zone, with the upper limit of normal at about 29 °C, and heat stress generally occurs at temperatures above 32 °C and 60% humidity [52,53]. In this experiment, the average rectal temperature ranged from 38.98 to 39.06 °C, indicating that the temperature was below the upper limit of the normal physiological range (38.1 to 39.2 °C) [9,54]. Kovács et al. [55], who studied heat stress in 7-week-old dairy calves, found a poor correlation between rectal temperature and meteorological parameters, especially in calves kept in shaded environments. Kovács et al. [55] have suggested that other physiological measures, such as skin temperature at the ear, respiratory rate, and heart rate, may be more appropriate indicators for assessing heat stress in young calves. Dairy calves housed indoors appear to breathe normally, between 24 and 36 times per minute [56]. Moore et al. [57] have also found that the respiratory rate of dairy calves increased by 2 breaths per minute with each 1 °C increase in indoor barn temperature. In this experiment, the average respiration rate ranged from 56.9 to 61.4 breaths per minute. This indicates that respiration rate exceeded the upper limit of the normal range, likely implying that the calves experienced heat stress during the experiment.

Nemati et al. [26] studied the effects of AH content and particle size in calves aged 0 to 10 weeks. They observed a three-way interaction between the content of AH, particle size, and the time calves consumed the starter feed. Calves offered a high content of AH with moderate particle size consumed the greatest amount of starer feed from week 8 to 10 of age, in contrast with what we observed in the current experiment. Previously, Karami et al. [23] reported that DMI of weaned dairy calves increased linearly when forage-to-concentrate ratio was decreased from 50:50 to 20:80. Aragona et al. [22] have also demonstrated that DMI was greater in calves fed restricted, long-chopped hay than those fed free-choice, long-chopped hay. These results may support the hypothesis that post-weaned calves may benefit from a limited amount of forage in their diets and providing greater amounts of forage can compromise their feed intake. Studies with ruminant animals indicate that increasing the forage-to-concentrate ratio limits feed intake via the physical distension of the gastrointestinal tract [58]. Our results indicate that feeding weaned calves the MF diet containing SPS AH may have reduced gut fill and minimized its inhibitory effect on DMI. It is possible that chopping AH into smaller sizes may have reduced heat production during heat stress and encouraged greater feed consumption. Further research is needed to ascertain whether reducing the length of forage has an impact on body heat production.

Regardless of forage particle size, we observed that calves receiving MF diets gained more weight and were heavier at the end of the experiment than those offered HF diets. The enhanced growth efficiency of calves fed MF diet might be related to the combined effects of greater DMI and improved nutrient digestion. In support of our findings, previous research has also shown that diets with higher starch content and higher NFC/NDF ratios promote growth performance in calves older than 8 weeks [59,60,61]. Aragona et al. [22] have also reported that feeding a high starch diet (45%) instead of a low starch diet (8%) results in increased feed intake, growth, and feed efficiency in calves at 2 to 4 months of age. The lower rectal temperature in calves fed MF diets could possibly be explained by their comparatively lower whole-body heat production and higher tissue energy retention, as evidenced by increased ADG and lower feed conversion ratio. Similarly, Reynolds et al. [62] have found that feeding a diet with 75% concentrate resulted in growing beef heifers producing less heat energy and retaining more tissue energy compared with those fed a diet containing 75% alfalfa.

In support of our findings that feeding MF vs. HF diet had a positive effect on several structural growth parameters, Aragona et al. [22] have reported that feeding a high starch, low fiber starter diet (51% starch) results in a 14% increase in hip width in young weaned calves (aged 2 to 4 months) compared with a low starch, high fiber starter diet (11% starch). Karami et al. [23] have also reported that heart girth, withers height, and hip eight at d 120 of calf age decrease linearly as forage-to-concentrate ratio increases from 20:80 to 50:50. Previous studies have shown that increasing ME intake can improve the body size of dairy calves [63]. Previous studies suggest that, as forage proportion in calf diet increased, ADG and empty body weight gain decreased [64,65,66]. In addition, as forage intake increased, the reticulorumen weight and length, as well as papillae density, decreased, which could affect the absorption of VFA and ultimately decrease growth and development [66].

Dairy producers have two primary goals for a cost-effective heifer-raising program: achieving earlier age at first calving and maximizing milk production during the first lactation [67]. Previous research suggests that promoting prepubertal growth and musculoskeletal development may potentially help in achieving these goals [68]. For example, accelerated growth in the first six months of a calf’s life has been shown to result in a lower age at first calving, which helps reduce heifer-raising costs [69]. Our preliminary data collected from this trial suggests the important role of the lower-fiber diet in promoting growth rate and in improving several skeletal growth measures of calves aged 3–6 months raised under challenging environmental conditions. Therefore, conducting long-term experiments is suggested in order to assess how this dietary factor might impact age at puberty and first calving, and the lactation performance of heifers as first-lactation cows.

Lower digestibility of DM and crude protein in calves receiving the HF than those receiving the MF diets could be explained by replacing lower-fiber, rapidly and extensively digestible components with medium-quality forage source with lower digestibility in HF diet. Feeding the HF diet may have exceeded the capacity of the undeveloped rumen and microbial community to digest nutrients, resulting in declined nutrient digestibility. Although AH particle size had no effect on digestibility of OM, DM, and crude protein, diets containing SPS vs. LPS AH increased NDF and ADF digestibility. Increased surface area of SPS AH may have resulted in increased digestibility of fibrous fractions by facilitating the attachment of ruminal cellulolytic bacteria to degrade cellulose and hemicellulose components [70]. Our results show that rumen pH tends to be higher in diets containing SPS vs. LPS AH, which is consistent with digestibility results. Lower ruminal pH may create an unfavorable environment for the optimal activity of cellulolytic bacteria in the rumen [71], compromising digestion of fibrous fractions.

Amat et al. [72] have reported that adult heifers fed diets with a lower forage-to-concentrate ratio had lower rumen pH than those fed diets with a higher forage-to-concentrate ratio. Although total rumen VFA production remained unaffected by dietary forage proportion, feeding HF diet increased eating and rumination time, which could promote salivary secretion. Previous studies have shown that forage supplementation in calf diets stimulates rumination activity [26,30], which is likely due to the necessary reduction in particle size of high-forage diets [73]. The longer duration of rumination in weaned calves fed HF compared with MF may be explained by the higher intake of peNDF, which primarily stimulates rumination activity [74].

Calves receiving HF vs. MF diets had greater acetate, but lower propionate molar proportions, resulting in acetate-to-propionate ratio to increase. Feeding high-fiber diets typically results in greater acetate proportions in the rumen [75]. In this experiment, greater acetate formation in HF-fed calves occurred at the expense of decreased propionate and valerate molar proportions. In agreement with our results, Nemati et al. [26] and Suarez-Mena et al. [76] have reported that feeding HF diets increased rumen acetate proportion, which could be related to higher pH and improved growth and activity of cellulolytic bacteria, resulting in higher acetate-to-propionate ratio [77]. Propionate is produced when NFC is fermented by amylolytic bacteria, while acetate results from the fermentation of structural carbohydrates by cellulolytic bacteria [78]. In support, heifers with body weights of 80 to 250 kg were fed diets with 12:88 vs. 30:70 forage-to-concentrate ratio, and the results suggest that the lower forage-to-concentrate ratio caused ruminal fermentation to become more amylolytic and proteolytic than cellulolytic, causing the propionate-to-acetate ratio and total VFA production to increase as heifer age progressed from 13 to 41 weeks [79]. Valerate may be used by cellulolytic microorganisms in the rumen [80], which could explain why its concentration decreased in HF-fed calves. In other words, the decrease in valerate concentration may be an indication of improved growth and activity of cellulolytic bacteria in HF-fed calves.

Although forage particle size did not affect the proportion of rumen VFA, the acetate-to-propionate ratio tended to be greater in calves receiving SPS vs. LPS AH. Nemati et al. [26] have also reported similar results. This response could be attributed to greater fiber digestibility, which shifts ruminal fermentation towards more acetate production, likely by stimulating cellulolytic microorganisms.

One of the key elements affecting the ruminating behavior in ruminants is the forage particle size. In this study, calves fed SPS AH spent longer eating and ruminating compared with calves fed LPS forage. Longer rumination time typically increases salivation during feeding, which further increases rumen pH, as seen in calves fed SPS AH. Studies in calves show contradictory results in relation to forage particle size and rumen pH. In contrast with our results, Nemati et al. [26] have reported that calves fed medium particle-sized AH (geometric mean = 3 mm) had higher rumen pH at both pre- and post-weaning periods (days 35 and 70) compared with those fed fine particle-sized AH (geometric mean = 1 mm). However, Mirzaei et al. [27] found that rumen pH was not different when calves were fed different particle sizes of AH (2.92 vs. 5.04 mm as geometrical mean). These discrepancies could be due to forage level, basal diet chemical composition or differences in calf stage of growth.

Our results indicate that the sorting behavior of growing calves against long particles may depend on dietary forage level as the greatest sorting against these particles occurred in calves fed the HF-SPS diet. Sorting against long particles of diet could also be an alleviating strategy for discomfort associated with the heat stress that occurred in the current study. As discussed earlier, greater forage proportion may increase heat increment in ruminants. Therefore, calves may prefer to consume less long particles in an effort to reduce the heat production that arises from physical and microbial digestion.

## 5. Conclusions

These findings provide valuable insight into the complex interplay between dietary fiber content, particle size, and performance of weaned calves (from approximate ages of 3 to 6 months). The findings suggest that the dietary fiber content has a greater impact on the performance of weaned calves under moderate heat stress, compared with the forage particle size. Weaned calves fed MF as compared with HF showed superior performance, as evidenced by greater feed intake and final body weight, improved nutrient digestion, and longer lying time, an indication of enhanced calf comfort. These results have practical implications for optimizing feed formulation for weaned calves in hot environments. Future research is needed to investigate the potential of manipulating dietary fiber content and forage particle size in order to optimize calf performance under different environmental conditions.

## Figures and Tables

**Figure 1 animals-14-00275-f001:**
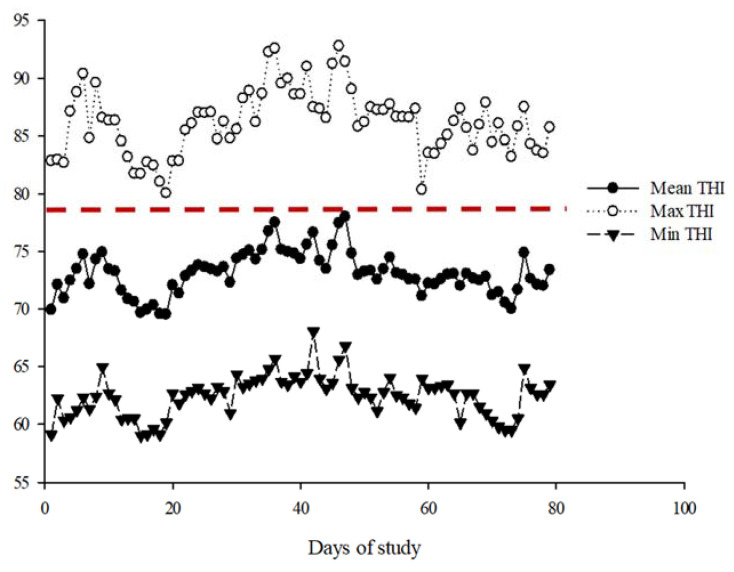
Temperature–humidity index (THI) during the experimental period. The cutoff point for heat stress is indicated by the dotted red line (THI ≥ 78).

**Figure 2 animals-14-00275-f002:**
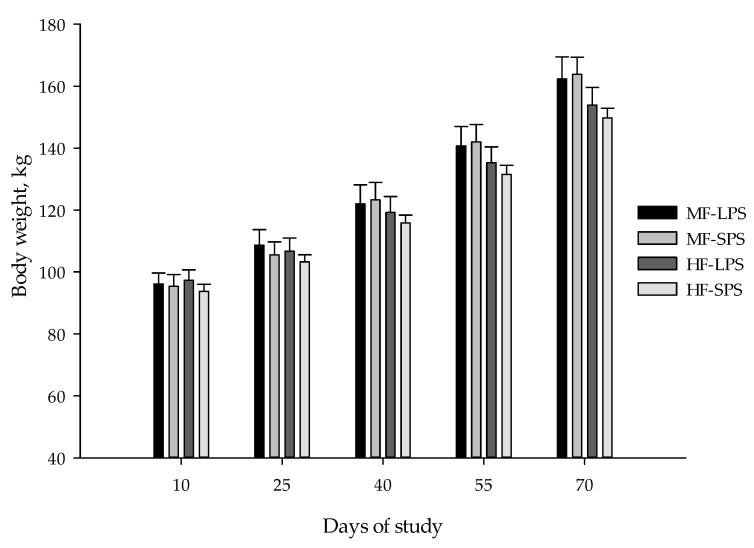
Calf body weight (means ± SE) at different time points. The dietary treatments were as follows: (1) high-fiber (HF) diets with forage-to-concentrate ratio of 40:60 with long particle-sized alfalfa hay (LPS; HF-LPS); (2) HF diets with short particle-sized alfalfa hay (SPS; HF-SPS); (3) moderate-fiber (MF) diets with forage-to-concentrate ratio of 22.5:77.5 with LPS alfalfa hay (MF-LPS); and (4) MF diets with SPS alfalfa hay (MF-SPS).

**Figure 3 animals-14-00275-f003:**
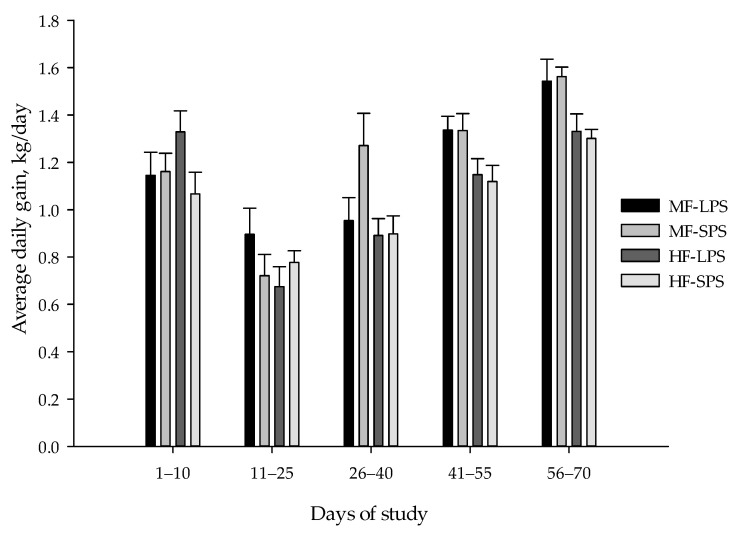
Calf average daily gain (means ± SE) at different time points. The dietary treatments were as follows: (1) high-fiber (HF) diets with forage-to-concentrate ratio of 40:60 with long particle-sized alfalfa hay (LPS; HF-LPS); (2) HF diets with short particle-sized alfalfa hay (SPS; HF-SPS); (3) moderate-fiber (MF) diets with forage-to-concentrate ratio of 22.5:77.5 with LPS alfalfa hay (MF-LPS); and (4) MF diets with SPS alfalfa hay (MF-SPS).

**Figure 4 animals-14-00275-f004:**
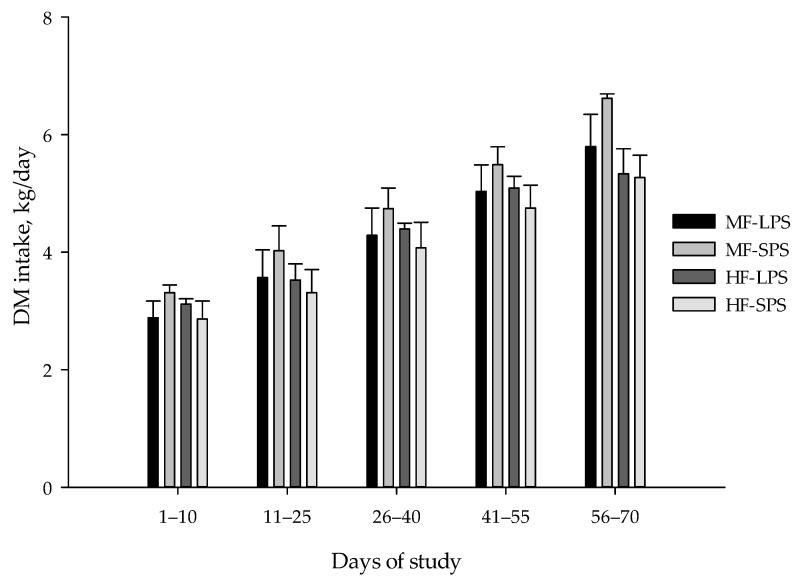
Calf DM intake (means ± SE) at different time points. The dietary treatments were as follows: (1) high-fiber (HF) diets with forage-to-concentrate ratio of 40:60 with long particle-sized alfalfa hay (LPS; HF-LPS); (2) HF diets with short particle-sized alfalfa hay (SPS; HF-SPS); (3) moderate-fiber (MF) diets with forage-to-concentrate ratio of 22.5:77.5 with LPS alfalfa hay (MF-LPS); and (4) MF diets with SPS alfalfa hay (MF-SPS).

**Figure 5 animals-14-00275-f005:**
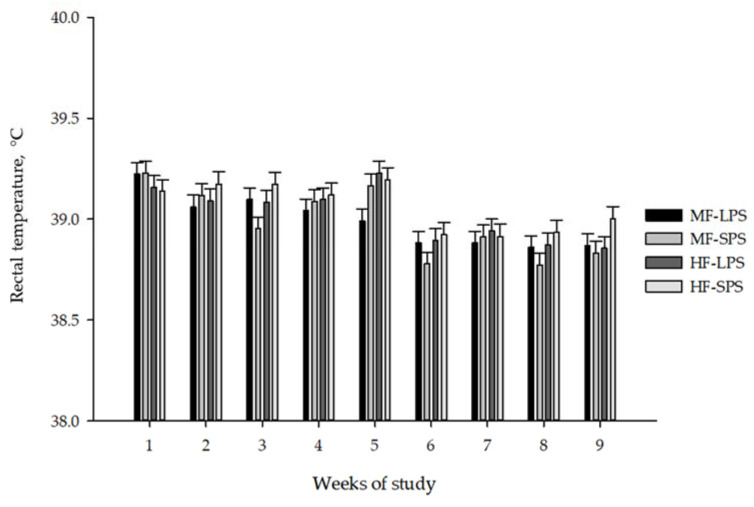
Calf rectal temperature (means ± SE) in weekly intervals. The dietary treatments were as follows: (1) high-fiber (HF) diets with forage-to-concentrate ratio of 40:60 with long particle-sized alfalfa hay (LPS; HF-LPS); (2) HF diets with short particle-sized alfalfa hay (SPS; HF-SPS); (3) moderate-fiber (MF) diets with forage-to-concentrate ratio of 22.5:77.5 with LPS alfalfa hay (MF-LPS); and (4) MF diets with SPS alfalfa hay (MF-SPS).

**Figure 6 animals-14-00275-f006:**
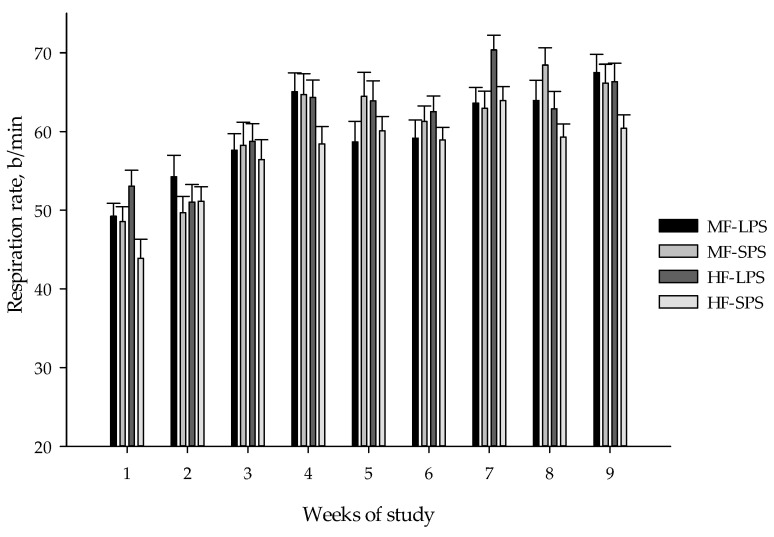
Calf respiration rate (means ± SE) in weekly intervals. The dietary treatments were as follows: (1) high-fiber (HF) diets with forage-to-concentrate ratio of 40:60 with long particle-sized alfalfa hay (LPS; HF-LPS); (2) HF diets with short particle-sized alfalfa hay (SPS; HF-SPS); (3) moderate-fiber (MF) diets with forage-to-concentrate ratio of 22.5:77.5 with LPS alfalfa hay (MF-LPS); and (4) MF diets with SPS alfalfa hay (MF-SPS).

## Data Availability

Data are available within the article.

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
