# Peer review of "Effects of Dietary Fiber Level and Forage Particle Size on Growth, Nutrient Digestion, Ruminal Fermentation, and Behavior of Weaned Holstein Calves under Heat Stress"

_animals, 2024, doi:10.3390/ani14020275_

Round 1

Reviewer 1 Report

Comments and Suggestions for Authors

In this manuscript, the authors investigated the complex interactions between particle size, dietary fiber content, and performance of weaned calves. Finally conclude,  moderate-fiber diets with shorter alfalfa hay improved weaned calves comfort under moderate heat stress, and the dietary fiber content had a greater impact on the performance of weaned calves compared to the forage particle size. This article has practical implications for optimising feed formulation for weaned calves under heat stress conditions. However, the manuscript readability could be much improved to better convey the importance of your study. With editing and some revisions, I feel that this manuscript will be suitable for publication.

1、 The Abstract part of the article has too many words and is not concise enough, please limit it to about 200 words.

2、 P in P<0.01 should be in italics, please check the entire article for consistent formatting.

3、 Line 140: What doesat 0600 and 1500 hmean? Please revise it.

4、 Line 147-148, Line 170: Same problems with Line 140.

5、 Table 3: There is a mistake tabulated on the line pef83. Please revise it in the table.

6、 Table 6:“Rumen fermentation parametersshould be on one line.

7、 What is the relationship between detection indicators such as Isobutyrate and Butyrate in Table 6 and rumen fermentation characteristics?

8、 A total of 76 references were cited for this article. Fifty-four references were from before 2015, of which 16 were from before 2000. For innovative articles, references should be nearly 3-5 years. Please update references.

Author Response

AU: Thank you for your valuable feedback on the original version. The manuscript has been revised according to your comments using the track change function in Word and yellow highlight.

1、 The Abstract part of the article has too many words and is not concise enough, please limit it to about 200 words.

AU: Many less important sentences were removed in the revised version, but we decided to keep the important information in the manuscript.

2、 P in P<0.01 should be in italics, please check the entire article for consistent formatting.

AU: Revised as instructed.

3、 Line 140: What does“at 0600 and 1500 h”mean? Please revise it.

AU: Revised as instructed. It was revised to 6:00 am and 3:00 pm. It was also revised in other parts to maintain consistency.

4、 Line 147-148, Line 170: Same problems with Line 140.

AU: Revised as instructed.

5、 Table 3: There is a mistake tabulated on the line pef83. Please revise it in the table.

AU: Corrected.

6、 Table 6:“Rumen fermentation parameters”should be on one line.

AU: Corrected.

7、 What is the relationship between detection indicators such as Isobutyrate and Butyrate in Table 6 and rumen fermentation characteristics?

AU: Thank you for your comment. Isobutyrate and Isovalerate are classified as branched-chain fatty acids in the rumen fluid and are required for most fiber-degrading microbes in the rumen. Acetate, propionate, butyrate, isobutyrate, valerate, and isovalerate are usually the main VFA form in the rumen fluid and are typically reported for VFA rumen profile in different ruminant species (calves, dairy cows, beef cattle, sheep, goat,…).

8. A total of 76 references were cited for this article. Fifty-four references were from before 2015, of which 16 were from before 2000. For innovative articles, references should be nearly 3-5 years. Please update references.

AU: We made an attempt to eliminate/update with new ones of certain outdated references from the manuscript. However, due to their significant relevance and necessity for methodology, we decided to retain some of them in the manuscript. This was done to ensure the integrity and completeness of the research methodology and its associated references. Also, according to the comments suggested by another reviewer, we added more sentences to the introduction and discussion chapters, and thus, more references were added to the revised version of this manuscript.

Reviewer 2 Report

Comments and Suggestions for Authors

The research, titled "Effects of dietary fiber level and forage particle size on growth, nutrient digestion, ruminal fermentation, and behavior of weaned Holstein calves under heat stress," aims to investigate the effects of feeding diets with varying fiber content and forage particle sizes on the performance, health, and welfare of Holstein dairy calves under heat-stress conditions. Despite this, the manuscript is of interest to Animals’ readership, given that calf welfare is considered highly relevant by consumers. Therefore, authors should emphasize this topic, especially as there is growing recognition that rearing is a key stage, laying the foundations for subsequent behavior, production, health, and welfare. Some useful information can be found in the reference 10.3390/vetsci10090554, which I recommend including in the introduction. Moreover, since these data pertain to animals not under heat stress conditions, they can also be used in the discussion section to comment on the data reported by the authors. However, there are several areas that could benefit from improvement:

In my opinion, the simple summary needs to be rewritten. According to the journal's instructions for authors (https://www.mdpi.com/journal/animals/instructions), the simple summary should contain a clear statement of the problem addressed, the aims and objectives, pertinent results, conclusions from the study, and how they will be valuable to society. It needs to be written for a lay audience.

I recommend rewriting the abstract to enhance its readability and suggest including more numerical data accompanied by significance levels.

The statistical analysis section needs expansion. I recommend including the model used for the analysis and its characteristics. I suggest reporting relevant references regarding some statistical methods (Shapiro Wilk and Levenes test) such as: 10.1186/s12917-022-03289-2.

I suggest dividing Table 1 into two parts: one for the chemical composition and one for the particle size distribution.

Furthermore, it would be beneficial to present the ingredients before the chemical composition of the diet.

Enrich the discussion by including the study's limitations and practical applications.

Author Response

The research, titled "Effects of dietary fiber level and forage particle size on growth, nutrient digestion, ruminal fermentation, and behavior of weaned Holstein calves under heat stress," aims to investigate the effects of feeding diets with varying fiber content and forage particle sizes on the performance, health, and welfare of Holstein dairy calves under heat-stress conditions. Despite this, the manuscript is of interest to Animals’ readership, given that calf welfare is considered highly relevant by consumers. Therefore, authors should emphasize this topic, especially as there is growing recognition that rearing is a key stage, laying the foundations for subsequent behavior, production, health, and welfare. Some useful information can be found in the reference 10.3390/vetsci10090554, which I recommend including in the introduction. Moreover, since these data pertain to animals not under heat stress conditions, they can also be used in the discussion section to comment on the data reported by the authors. However, there are several areas that could benefit from improvement:

In my opinion, the simple summary needs to be rewritten. According to the journal's instructions for authors (https://www.mdpi.com/journal/animals/instructions), the simple summary should contain a clear statement of the problem addressed, the aims and objectives, pertinent results, conclusions from the study, and how they will be valuable to society. It needs to be written for a lay audience.

AU: Thank you very much for the constructive feedback on our manuscript. the manuscript has been revised according to the comments as trackable using the track change function in Word.

We agree with the comment and the simple summary was thoroughly revised to make it fit for a lay audience as follows:

Simple Summary: Global temperatures are on the rise, and this poses a significant threat to animal farming. However, there are certain nutritional strategies that can help mitigate the harmful effects of heat stress, particularly for growing heifer calves, as they play a crucial role in the future profitability and sustainability of a dairy farm. In this study, growing calves experiencing heat stress consumed more feed and digested nutrients more efficiently when offered diets with moderate content of dietary fiber and shorter alfalfa hay. Feeding a diet with a moderate level of fiber resulted in better growth, lower rectal temperature, and longer lying behavior, an indication of better animal comfort. Our findings highlight the importance of dietary fiber level and forage particle size in shaping calf performance and behavior under challenging environmental conditions”.

This sentence was also added to the introduction chapter:

Appropriate management strategies during pre- and post-weaning phases are crucial, laying the foundations for long-term productivity, behavior, and health of heifer calves, and ultimately contributing to a successful heifer-rearing program [1, 2].

  1. Khan M, Weary D, Von Keyserlingk M. Invited review: Effects of milk ration on solid feed intake, weaning, and performance in dairy heifers. Journal of Dairy Science. 2011;94:1071-81.
  2. Cavallini D, Raspa F, Marliani G, Nannoni E, Martelli G, Sardi L, et al. Growth performance and feed intake assessment of Italian Holstein calves fed a hay-based total mixed ration: preliminary steps towards a prediction model. Veterinary Sciences 2023;10:554.

I recommend rewriting the abstract to enhance its readability and suggest including more numerical data accompanied by significance levels.

AU: The Abstract was also revised extensively, removing less important data while keeping the more important ones. We also avoided reporting tendency effects in the revised version.

The statistical analysis section needs expansion. I recommend including the model used for the analysis and its characteristics. I suggest reporting relevant references regarding some statistical methods (Shapiro Wilk and Levenes test) such as: 10.1186/s12917-022-03289-2.

AU: We have revised the statistical analysis section as per the instructions provided. The changes aim to clarify the effects of pen or individual animal on the assessed outcomes. The characteristics of the model were improved specifying whether the pen or individual calves are specified in the model. The interaction was also specified, with those measurements repeated over time.

Appropriate reference was also added regarding the normality test as recommended.

I suggest dividing Table 1 into two parts: one for the chemical composition and one for the particle size distribution. Furthermore, it would be beneficial to present the ingredients before the chemical composition of the diet.

AU: As suggested, Table 1 was divided into 2 parts: Table 1 for the chemical composition and Table 2 for the particle size distribution. The necessary changes were made in the manuscript.

Table 1 displays only the chemical composition of the forages used in the experimental diet formulation, namely corn silage and alfalfa hay. Therefore, it is related to Table 2, which contains the diet tables. In Table 2, the ingredients have been listed first, followed by their respective chemical compositions (Table 3).

Enrich the discussion by including the study's limitations and practical applications.

AU: We agree with the comment and this sentence was incorporated into to the discussion chapter.

Dairy producers have two primary goals for a cost-effective heifer-raising program: achieving earlier age at first calving and maximizing milk production during the first lactation [67]. Previous research suggests that promoting prepubertal growth and musculoskeletal development may potentially help in achieving these goals [68]. For example, accelerated growth in the first six months of a calf life has resulted in a lower age at first calving, which helps reduce heifer-raising costs [69]. Our preliminary data collected from this trial suggests the important role of the lower-fiber diet in promoting growth rate and several skeletal growth measures of calves aged 3-6 months raised under challenging environmental conditions. Therefore, conducting long-term experiments is suggested to assess how this dietary factor might impact age at puberty and first calving, and the lactation performance of heifers as first-lactation cows.

References used:

[1] Salte R, Storli K, Wærp H, Sommerseth J, Prestl E, Volden H, et al. Designing a replacement heifer rearing strategy: Effects of growth profile on performance of Norwegian Red heifers and cows. Journal of dairy science. 2020;103:10835-49.

[2] Nor NM, Steeneveld W, Van Werven T, Mourits M, Hogeveen H. First-calving age and first-lactation milk production on Dutch dairy farms. Journal of Dairy Science. 2013;96:981-92.

[3] Brickell J, Bourne N, McGowan M, Wathes D. Effect of growth and development during the rearing period on the subsequent fertility of nulliparous Holstein-Friesian heifers. Theriogenology. 2009;72:408-16.

Reviewer 3 Report

Comments and Suggestions for Authors

A well-written manuscript. I have offered some suggestions on the edited draft to help correct some errors and improve the quality of the manuscript.

Author Response

AU: Thank you for your valuable feedback on the original version. We confirm that the revised manuscript has been thoroughly and carefully revised according to the specific comments provided in the original version of the manuscript using the track change function in Word and yellow highlight. 

Round 2

Reviewer 2 Report

Comments and Suggestions for Authors

the paper improved a lot